# Insulin Resistance and Acne: The Role of Metformin as Alternative Therapy in Men

**DOI:** 10.3390/ph16010027

**Published:** 2022-12-26

**Authors:** Aikaterini Andreadi, Saverio Muscoli, Rojin Tajmir, Marco Meloni, Alessandro Minasi, Carolina Muscoli, Sara Ilari, Vincenzo Mollace, David Della Morte, Alfonso Bellia, Elena Campione, Nicola Di Daniele, Davide Lauro

**Affiliations:** 1Department of Systems Medicine, Section of Endocrinology and Metabolic Diseases, University of Rome Tor Vergata, 00133 Rome, Italy; 2Division of Endocrinology and Diabetology, Fondazione Policlinico Tor Vergata, 00133 Rome, Italy; 3Division of Cardiology, Fondazione Policlinico Tor Vergata, 00133 Rome, Italy; 4Department of Health Science, University of Magna Graecia, 88100 Catanzaro, Italy; 5Division of Internal Medicine—Hypertension, Department of Medical Sciences, Fondazione Policlinico “Tor Vergata”, 00133 Rome, Italy; 6Department of Neurology, Evelyn F. McKnight Brain Institute, University of Miami Miller School of Medicine, 1120 NW 14th St., Miami, FL 33136, USA; 7Unit of Dermatology, Fondazione Policlinico Tor Vergata, 00133 Rome, Italy

**Keywords:** insulin resistance, metformin, acne, drug efficacy evaluation

## Abstract

The association between acne and insulin resistance has not been investigated as thoroughly in males as it has been in women, despite the fact that in adult men, acne prevalence has grown. On the face, sebaceous glands produce and secrete sebum, which lubricates the skin and protects it from friction. Metformin, an insulin-sensitizing medication, may modify the association between acne vulgaris and insulin resistance (IR). Individuals with IR, metabolic syndrome or with impaired glucose tolerance are sometimes treated ‘off label’ with Metformin. In these conditions, IR may be a leading factor in the pathogenesis of acne, and in men, Metformin treatment may reduce the Global Acne Grading System (GAGS) score by enhancing insulin sensitivity. However, additional clinical studies are required to corroborate these assumptions.

## 1. Introduction

Acne vulgaris is a chronic inflammatory state with papules, pustules, nodules and cysts in the pilosebaceous units. This is one of the most common skin disorders worldwide, affecting around 22% of the population and adolescents in particular [1,2]. Recently, the incidence and prevalence of acne disease has increased in adult patients [3].

The sebaceous units are holocrine glands that are located all over the body’s surface, except for the feet, palms, soles and dorsal surfaces. They are especially prevalent and dense on the face and scalp, which are the origins of acne formation [4].

The sebaceous glands usually produce and release sebum. The sebaceous lipids are primarily non-polar lipids comprising triglycerides and wax esters. In addition, there are also cholesterol, squalene and fatty acid breakdown products. Sebum production lubricates the skin and protects it from friction (Figure 1) [5]. Furthermore, the sebaceous glands distribute antioxidants into and onto the skin and possesses a natural light-protective function. Moreover, the sebaceous gland lipids feature pro-and anti-inflammatory properties with an antimicrobial action and modulating immune cells [5,6].

It is common knowledge that androgens regulate how male and female reproductive systems develop and function [7,8]. The main androgen in the blood is testosterone, which is primarily produced by Leydig cells in the male testis. Although the adrenal glands can also secrete testosterone, the Leydig cells in the male testis are the primary circulating androgens. With the help of 5α-reductases, testosterone can be converted into the more potent androgen 5α-dihydrotestosterone (DHT). Both testosterone and DHT have the ability to bind to the nucleus-localized androgen receptor (AR) [9]. However, DHT is ten times more affine than testosterone. Even though the skin doesn’t produce significant amounts of androgens, sebocytes, sweat glands and dermal papilla cells can modify the androgenic pro-hormone dehydroepiandrosterone (DHEA) and androstenedione into testosterone and DHT. These potent androgens subsequently regulate the cutaneous physiology in an intracrine or paracrine way [10].

Androgens have a crucial function and contribute to defining the size of the sebaceous gland. Which include the necessary enzymes for the synthesis and metabolism of androgen from cholesterol to 5α-dihydrotestosterone (DHT). Androgen levels are one of the leading causes for the onset and severity of acne vulgaris by increasing the activity of sebaceous glands and sebum production and promoting the growth of keratinocytes.

Androgen levels can increase lipid synthesis, sebocyte proliferation and differentiation. After androgens bind to the AR, the phosphorylation of the mammalian target of rapamycin (mTOR) increases [11]. It was found that the cytoplasmic and nuclear expression of mTOR was elevated in inflammatory sebaceous glands of acne lesions relative to non-lesioned skin. The mTOR is the catalytic core of the mammalian target of rapamycin complex 1 (mTORC1 complex), which stimulates lipogenesis by activating the sterol regulatory element-binding protein-1 (SREBP-1) [11].

The endogenous Wnt/-catenin signaling pathway was negatively regulated by androgens. Consequently, the expression of the Wnt/-catenin target genes such as c-MYC increases induced sebocyte differentiation. The sebocytes undergoing differentiation include an abundant nuclear AR and peroxisome proliferator-activated receptors (PPARs). During this process, the lipids will gradually accumulate until the sebocytes are sufficiently developed to secrete their contents via holocrine secretion. This will take place once the sebocytes have reached the appropriate level of development [12,13].

In addition, the sebaceous glands contain most of the steroidogenic enzymes for the conversion of DHEA/DHEAS (DHEA sulfate) to testosterone and DHT [14,15]. There are three isoforms of 5α-reductase, and their expression patterns vary among species and tissues [16]. Type I 5α-reductase is mainly expressed in sebocytes, keratinocytes and dermal fibroblasts Type II 5α-reductase is mainly detected in seminal vesicles, epididymis, prostate and fibroblasts from adult genital skin and in the inner root sheath of the hair follicle. The newly discovered type III 5α-reductase is detected in prostate cancer and sebocyte cell lines [17,18]. In addition to steroidogenic activity, type III 5α-reductase is critically involved in *N*-linked glycosylation [19]. Interestingly, the sebum production rate in patients with a type II 5a-reductase deficiency was similar to that in normal men [20], suggesting that DHT produced locally by type I 5α-reductase enhances the sebum production. However, in clinical and in vitro studies, selective inhibitors of type I 5α-reductase did not significantly reduce the sebum production or improve acne vulgaris [21,22], suggesting that the suppression of 5α-reductase alone is not sufficient to improve acne. There are several possibilities to explain this. First, the suppression of a single 5α-reductase type might not be sufficient to completely block the DHT synthesis because there are redundancies between the different 5α-reductase types and the sebaceous glands are sensitive to even minute amounts of DHT. Second, the newly discovered type III 5α-reductase may play a more important role in regulating sebum production. Third, testosterone and DHT have different effects on the activation of AR [23], suggesting that testosterone rather than DHT may be a more important regulator of sebum production. Moreover, the involvement of AR coregulators could compensate for the deficit in DHT production.

During puberty, physiologic insulin resistance (IR) generates hyperinsulinemia, leading to increased androgen levels, and hyperinsulinemia and hyperandrogenemia may promote the onset of acne [24]. IR and the associated hyperinsulinemia promotes a pituitary LH release, a rise in testosterone production and an inhibiting SHBG (sex hormone binding globulin) synthesis, leading to high levels of free testosterone (FT). This mechanism may also be present in men, partially explaining the correlation between hyperandrogenism and IR [25].

Higher insulin levels enhance the concentration of the insulin-like growth factor 1 (IGF-1) and, consequently, reduce the insulin-like growth factor binding protein-3 (IGFBP-3). The IGF-1 may stimulate the mean face sebum excretion and increase the dihydrotestosterone and dehydroepiandrosterone sulfate serum levels, but it also induces an increase in sebocyte proliferation [24].

Moreover, hyperinsulinemia promotes epidermal growth factors and the transforming growth factor β, which raise plasma levels of non-esterified fatty acids, causing inflammation and possibly the colonization of the intrafollicular duct by Cutibacterium acnes and the development of acne vulgaris [26]. Insulin resistance is fundamental to the onset of metabolic disorders that characterize the metabolic syndrome.

## 2. Insulin Resistance, IGF-1 and IGFBP-3

Recent research indicates that diets with a high glycemic load promote the acne vulgaris disease by increasing the IGF-1 levels. People who have acne and consume diets with a low glycemic load have fewer acne lesions compared to people who consume diets with a high glycemic load. This is because diets with a low glycemic load include fewer carbohydrates. This finding has been demonstrated by several researchers who investigated the impact of the glycemic index of various diets and glycemic loads in individuals suffering from acne [27]. Smith et al. demonstrated that a low glycemic regimen for 12 weeks significantly reduced the serum IGF1 concentrations and alleviated acne disease [28]. A dietary change as a low glycemic regimen also booted the nuclear concentration of the factor forkhead box O1 (FOXO1), normalizing the transcription of acne-related genes.

Hyperglycemia generated by food intake with a high glycemic index increases the insulin secretion of pancreatic beta cells [29] high insulin blood concentrations decrease the hematic glucose levels (Figure 2) [30]. Additionally, the stress in response to glucose reduction and eventual hypoglycemia may stimulate the adrenal gland androgens production with an increased hepatic glucose production to restore normal blood glucose levels [31]. Furthermore, low blood glucose levels increase the need for food and induce appetite. These components start generating a vicious-cycle that is perpetually repeated, generating high levels of food intake and calories thereby increasing the IR levels. The IR associated with hyperinsulinemia and hyperglycemia generates different pathologic mechanisms and, among others, increases the cellular GLUT4 endocytosis leading to a decreased uptake of glucose. This defect can be reversed though Metformin treatment [32].

Chronic IR triggers a cascade of events that promotes the development of the sebaceous gland tissue and holocrine secretion by increasing the free insulin growth factor-1 (IGF-1) and decreasing the insulin growth factor binding protein-3 (IGFBP-3) [31]. The free IGF-1, a potent mitogen, promotes acne development by upregulating hyperkeratinization. The reduction of the IGFBP-3 levels enhances the availability of the free IGF-1, which upregulates cell proliferation. The IGF-1 activates 5a-reductase activity [33], and the IGFBP-3, a nuclear retinoid X receptor (RXR) ligand, amplifies the RXR homodimer-mediated signaling [34]. Therefore, the decreased IGFBP-3 may diminish the ability of the endogenous retinoids to activate the genes that regulate follicular cell proliferation in the skin. In addition, hyperinsulinemia raises the secretion of the epidermal growth factors (EGFs) and the transforming growth factor (TGF)-alpha, increasing the non-esterified fatty acid concentrations. While higher fatty acids levels reduceIGFBP-1, 2 and 3 production, the production of IGF-1 is increased [34].

## 3. Metformin and Acne

Topical retinoids, azelaic acid and benzoyl peroxide are some of the acne treatments suggested by current European standards. Other therapies for acne include systemic medications and topical fixed-dose combinations. Oral isotretinoin and systemic antibiotics are the only two systemic therapies available and either a high or medium recommendation is given for their use. Oral antiandrogens are an alternative treatment for female patients where there are no contraindications for recommending this course of action.

Some substances can inhibit the synthesis of androgen. For instance, spironolactone can decrease testosterone production and competitively prevent androgens, especially DHT and testosterone, from binding to the skin’s androgen receptors. The inhibition of 5-α-reductase and the elevation of the sex hormone-binding globulin (SHBG) production may be another way. The anti-androgenic features of the combination of the oral contraceptive tablets contribute to their acne-fighting efficacy.

In addition to inhibiting 5-α-reductase activity, blocking androgen receptors and increasing the SHBG’s capacity to bind free testosterone, contraceptive medication can also lower the amount of androgen synthesized by the ovaries.

As antagonists for the sebocyte receptors, several compounds exist. Flutamide, a selective androgen receptor blocker generally used to treat prostate cancer, can also be used to cure acne, according to recent findings. Due to the fact that the Food and Drug Administration (FDA) has not authorized this medication as an acne treatment, it is not suggested. It should be taken only when the potential benefit outweighs the risk. Additionally, the PPAR modifier *N*-Acetyl-GED-0507-34-LEVO and melanocortin receptor (MCR) antagonists block the sebocyte receptors [35]. Both of them reduce the creation of sebum. Several medications inhibit lipid production via Akt/FoxO1/mTOR or AMPK signaling while lipid production is inhibited by a number of pharmaceuticals. Epigallocatechin-3-gallate (EGCG) suppresses the IGF-induced lipogenesis in SZ95 sebocytes by reducing the mTOR and S6 ribosomal protein levels [36]. Both are important elements for the downstream components of the Akt pathway. EGCG suppresses sebum production by stimulating the AMPK-SREBP-1 signaling pathway [36].

The FDA has authorized oral isotretinoin as an effective treatment for the majority of individuals with severe, resistant acne. Since it upregulates the expression of the nuclear FoxO1 and FoxO3 proteins, it causes sebocyte death to reduce the sebum production. Oral isotretinoin may also be utilized for the treatment of moderate acne with a tendency to scar, a considerable psychosocial impairment and a high risk of recurrence. Metformin is an indirect inhibitor of mTORC1 because it activates the AMPK pathway, which is a negative regulator of mTORC1 [37]. In the polycystic ovarian syndrome, metformin has demonstrated beneficial results in the treatment of acne. Other drugs with comparable therapeutic effects block the lipid synthesis enzymes. For instance, olumacostat glasaretil, an inhibitor of acetyl coenzyme A carboxylase (ACC), reduces triacylglycerol levels in sebocytes [38]. XEN103,6-[4-(5-fluoro-2-trifluoromethylbenzoyl)-piperazin-1-yl] inhibits stearoyl-CoA desaturase (SCD), an enzyme regulated by SREBP-1, and pyridazine-3-carboxylic acid (2-cyclopropylethyl) amide decreases lipid levels in sebocytes as well as the size and quantity of the sebaceous glands [39]. In addition to inhibiting SCD, XEN103 reduces the androgen-induced expression of SCD.

Metformin, a biguanide hypoglycemic agent that is safe and effective for treating acne in women with polycystic ovary syndrome (PCOS), has evidence of improving insulin resistance, hyperandrogenism, hyperlipidemia, cardiovascular health, quality of life, psychological well-being and overall health outcomes [37]. It is an insulin-sensitizing medication that may play a role in the interplay between acne and IR. In addition, patients with acne disease have enhanced probabilities of developing IR, but as reported previously, IR may play a role in the pathogenesis of this disease. A higher risk of metabolic and vascular damage has been linked to chronic inflammatory skin diseases. Metformin treatment benefits multiple facets of the metabolic syndrome, including hyperglycemia, obesity and liver dysfunction. Additionally, it has been demonstrated that Metformin significantly decreases the risk of cardiovascular disease [1,40], and evidence suggests that the Mediterranean and vegetarian diets are connected with several health benefits, including a lower risk of cardiovascular disease. These benefits are associated with plant-based diets in general [41]. Metformin also has anti-inflammatory benefits for a variety of cell types. These effects include a reduction in nitric oxide production, pro-inflammatory cytokines (IL-1, IL-6) and TNF-α, whose mechanisms are more complex [42]. Furthermore, Metformin can exert an anti-inflammatory action by inhibiting the activation of nuclear factor kappa B (NF-κB). These effects may reduce the pro-inflammatory processes, which are already activated in macrophages and are also present in cutaneous sebocytes, as reported in Figure 1. Metformin’s anti-inflammatory action is mainly triggered by the activation of AMP-activated protein kinase (AMPK), suppressing the NF-κB pathway. Furthermore, Metformin can suppress mTORC1 activity and may lead to an improvement in acne development and appearance [37]. Finally, Metformin mediated the suppression of the IGF-1 serum levels and retinoid-induced up-regulation of the IGFBP-3 expression to down-regulate the IGF1R signal transduction thereby exerting the inhibitory effects on the synthesis and conversion of the androgens and AR ligand binding [43]. This effect may play an important role in the cure of acne vulgaris.

## 4. Discussion

Several studies have already evaluated the therapeutic role of Metformin in female patients affected by acne and insulin resistance [30]. The most common cause of hyperandrogenemia in women is polycystic ovary syndrome (PCOS), which usually manifests as acne, hirsutism and menstrual irregularities. Insulin resistance and raised plasma levels of insulin are responsible for the high androgen concentration in patients with PCOS. However, literature studying metformin’s effects on the severity of acne in patients with PCOS is well documented compared to men [44]. The search for publications was conducted in September 2022. Candidate studies were retrieved from the PubMed database (http://www.ncbi.nlm.nih.gov, accessed on 5 September 2022) using the keywords acne, insulin resistance, metformin and men. All papers written in English and published during the period 2002–2021 were reviewed. The reference list of all retrieved articles was also reviewed to identify any additional eligible studies that were not indexed in the above database. With these studies, our contribution identifies how underestimated acne in men is in previous research and the use of metformin in men, as affected in women. The male correlation between acne and insulin resistance has not been extensively studied. Nagpal et al. designed a trial that enrolled 100 patients with acne and 100 patients without, who were in the same age group. The purpose of this cross-sectional study was to investigate the frequency of insulin resistance and metabolic syndrome in 20-year-old and older male patients with acne disease (Table 1) [45]. The population was divided into four groups depending on the acne severity: mild, moderate, severe and very severe. The mean weight and body mass index (BMI) between the control and study subjects were comparable, even if the population with severe acne disease had significantly higher levels of weight and BMI compared to people with a mild appearance of acne disease. Homeostasis model assessment-insulin resistance (HOMA-IR), an indirect index to measure insulin resistance based on the fasting glucose and insulin levels, was used and the values were comparable between the different acne severity groups. Moreover, HOMA-IR values were considerably higher in the patients than in the control subjects. Young males with acne had an increased chance of developing a rise in the mean values of HOMA-IR and fasting plasma glucose. For increased levels of IR, young males also showed higher mean values of systolic blood pressure (SBP) and diastolic blood pressure (DBP) compared to the control. In people with more severe grades of acne disease, body weight IR may be higher and could be associated with hyperinsulinemia, leading to an increased risk for the development of Type 2 Diabetes Mellitus [45].

Robinson et al. conducted the first clinical trial investigating the effectiveness and safety of Metformin as an adjuvant therapy in healthy subjects with acne disease, regardless of the patient’s sex, BMI or insulin resistance (Table 1) [37]. This trial included 84 patients. Of this sample size, 42 were treated with Metformin (850 mg) + tetracycline + Benzoyl peroxide (BPO) while 42 were treated with tetracycline + BPO as a control group. The number of male patients in the Metformin and control groups was 15 and 10, respectively. The results from this study suggested that hyperglycemia worsens acne severity since the fasting blood glucose was positively correlated with total acne lesion counts. These data suggested an improvement in clinical acne disease with the use of Metformin as an adjunct therapy in male and female patients with moderate to severe acne, regardless of their BMI status [37].

Fabbrocini et al. designed a study in which 20 male patients with altered metabolic profiles (e.g., impaired fasting glucose, high total cholesterol, LDL, low HDL and higher values of waist circumference and BMI) were enrolled (Table 1) [46]. This population was divided into two groups of 10 individuals, with Group A receiving Metformin 500 mg twice daily in addition to a hypocaloric diet (1500–2000 kcal) and standard acne treatment. Group B continued with the standard acne treatment they had used before the trial. The Global Acne Grading System (GAGS) was used to define the severity of acne disease. The Global Acne Grading System (GAGS) is a clinical instrument initially proposed by Doshi et al. in 1997. Its purpose is to evaluate the severity of acne vulgaris. Lesions caused by acne can be subdivided into four categories in increasing order of severity as follows: comedones, often known as whiteheads and blackheads; papules, irritating lumps that are no larger than one centimeter in size; pustules, papules that have a pus-filled head (pusheads); and nodules and cysts, bumps that are at least one centimeter in size. Cysts are hollow and filled with fluid, whereas nodules are solid [47].

The GAGS considers six different places on the face, chest and upper back, assigning a factor to each region based on the surface area, distribution and density of the pilosebaceous units. The hairline, jawline and earlobes define the facial borders. Due to the significance of the chest and upper back in determining the severity of acne and selecting the appropriate therapy, these areas have been considered (Figure 3). Each of the six places receives its score, based on a scale from 0 to 4 and is determined by the lesion that is considered to be the most severe within that area. The culmination of the score in each region constitutes the global score. The global score is calculated by adding up all of the local scores, which correspond to moderate to severe acne on our scale (Table 2) [47].

After six months of Metformin treatment, Group A had a significant reduction in the GAGS score from 25.1 ± 8.9 to 14.1 ± 10.4, with a decrease in glucose values, 120 min after a 75 g Oral Glucose Tolerance Test (OGTT), without any mild or severe side effects. Group B did not report any significant improvement in GAGS (24.9 ± 7.6 to 19.4 ± 7.4) [46].

A study by Del Prete et al. examined the relationship between insulin resistance and other metabolic disorders in male acne patients [48]. This study comprised 22 boys with acne resistance to treatment, aged from 15 to 26 years, along with 22 healthy males of a comparable age who participated as the control group (Table 3). The results showed that the levels of total cholesterol, triglycerides, serum IGF-1 and testosterone were comparable between the two groups. In contrast to the control group, the patients with resistant acne disease had a higher BMI, blood insulin and glucose concentrations and lower HDL cholesterol. In all the patients with resistant acne disease, the androgenic profile was reported as normal [48]. These data suggested that the chief influence of high insulin and glucose levels, low HDL-cholesterol and adiposity (essentially metabolic syndrome) on acne disease in males was different from females, which is chiefly influenced by androgen production and action. These findings highlighted a strong relationship in the etiopathogenesis of acne disease between inflammation, insulin resistance and cholesterol levels, at least in males.

## 5. Conclusions

A low-carbohydrate diet combined with conventional medication may improve the clinical course of acne disease and insulin sensitivity in young men who are resistant to dermatological drug therapy and have high levels of IR and metabolic syndrome features. Metformin may reduce the GAGS score, improving insulin resistance. Further clinical studies are needed to confirm these conclusions and to amplify and validate the use of Metformin in male patients affected by acne and insulin resistance.

## Figures and Tables

**Figure 1 pharmaceuticals-16-00027-f001:**
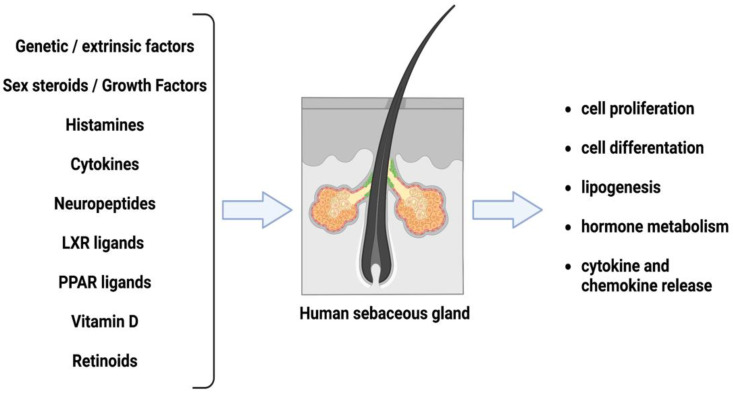
Regulation of the biological function of human sebaceous gland cells. The biological process of sebocyte holocrine secretion is regulated by several factors, including the ligands of the expressed receptors, such as androgens and estrogens, peroxisome proliferator-activated receptors (PPAR) ligands, neuropeptides, liver X receptor ligands (LXR), histamines, retinoids and vitamin D. The ligand-receptor complexes activate the pathways involving cell proliferation, differentiation, lipogenesis, hormone metabolism and cytokine and chemokine release.

**Figure 2 pharmaceuticals-16-00027-f002:**
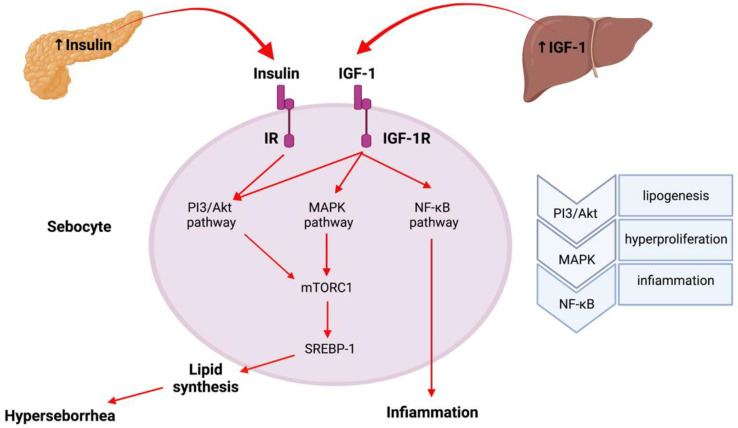
Insulin–IGF-1 action at the sebaceous gland level: Insulin and the IGF-1 induce lipid synthesis by increasing the SREBP-1 expression via the PI3K/Akt/FoxO1/mTORC1 pathway and MAPK pathway. The IGF-1 may induce the proinflammatory cytokine expression in sebocytes, activating the NF-κB activity. (AMPK: AMP-activated protein kinase, NF-κB: nuclear factor kappa B, SREBP-1: sterol regulatory element-binding protein 1, PI3: phosphatidylinositol-3-kinase, Akt: protein kinase B, mTORC1: mammalian target of rapamycin complex 1).

**Figure 3 pharmaceuticals-16-00027-f003:**
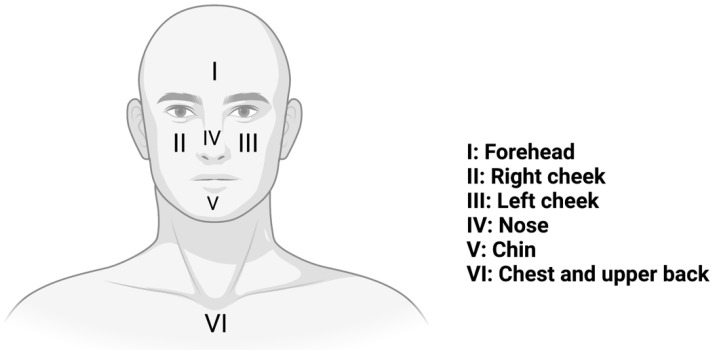
The six locations (I–VI) of the Global Acne Grading System (GAGS).

**Table 1 pharmaceuticals-16-00027-t001:** Studies evaluating the efficacy and safety of Metformin in male patients affected by acne (GAGS: Global Acne Grading System, HOMA-IR: homeostasis model assessment-insulin resistance).

Study	Population		Mean Age—Year			Metformin		GAGS		HOMA-IR			BMI kg/m^2^			Plasma Insulin, μIU/mL			Total Cholesterol mg/dL		HDL Cholesterol mg/dL	
	Case	Control	Case	Control	*p* Value	Case	Control	Case	Control	Case	Control	*p* Value	Case	Control	*p* Value	Case	Control	*p* Value	Case	Control	Case	Control
Robinson et al., 2019 [37]	42 (35.7% male)	42 (23.8% male)	22.6	23.2	0.501	Yes		N/A	N/A	1.6	1.8	0.295	21.6	22.6	0.303	7.4	8.2	0.468	N/A	N/A	N/A	N/A
Fabbroccini et al. 2016 [46]	10	10	19.5	19.5	N/A	Yes		14 ± 10.4	19.4 ± 7.418	1.5 ± 0.1	1.5 ± 0.8	N/A	22.9 ± 3.5	24.1 ± 2.5	N/A	9.6 ± 7.5	10.4 ± 1.6	N/A	165 ± 4.8	166 ± 14.5	49.5 ± 0.5	50 ± 0.4

**Table 2 pharmaceuticals-16-00027-t002:** The GAGS: Global Acne Grading System explanation [47].

GAGS (Global acne grading system)	GRADE
Location	Score	No lesion	0
I Forehead	2	Comedones	1
II Right cheek	2	Papules	2
III Left cheek	1	Pustules	3
IV Nose	1	Nodules/cysts	4
V Chin	1	Nodules/cysts	4
VI Chest and upper back	3		
Local score: multiplicate factor × GRADE (0–4)	
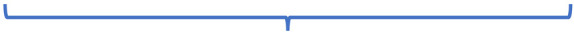
	Global score	
	0	none	
	1–18	mild	
	19–30	Moderate	
	31–38	severe	
	>39	Very severe	

**Table 3 pharmaceuticals-16-00027-t003:** Studies evaluating the correlation between insulin resistance and acne in male patients. (GAGS: Global Acne Grading System, HOMA-IR: homeostasis model assessment-insulin resistance).

Study	Population		Mean Age—Years			Metformin		GAGS		HOMA-IR			BMI kg/m^2^			Insulin, μIU/mL			Total Cholesterol mg/dL		HDL Cholesterol mg/dL	
	Case	Control	Case	Control	*p* Value	Case	Control	Case	Control	Case	Control	*p* Value	Case	Control	*p* Value	Case	Control	*p* Value	Case	Control	Case	Control
Nagpal et al., 2016 [45]	100	100	22.7	23.7	0.06	No		N/A		2	1.7	0.49	22.9	23.4	0.37	9.2	7.8	0.22	N/A	N/A	42.5	40.8
Del Prete et al., 2012 [48]	22	22	18.6	20.2	0.003	No		N/A		1.7	1.1	0.016	24	20.1	0.003	10.6	5.5	0.01	N/A	N/A	46.5	57.3

## Data Availability

Data sharing not applicable.

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
