# Peer review of "Insulin Resistance and Acne: The Role of Metformin as Alternative Therapy in Men"

_pharmaceuticals, 2022, doi:10.3390/ph16010027_

Round 1

Reviewer 1 Report

In the present narrative review, Aikaterini Andreadi and co-workers discussed about the effects of Metformin, an insulin-sensitizing medication, on the pathological association between acne vulgaris and insulin resistance. They concluded that Metformin treatment may reduce the Global Acne Grading System score by enhancing insulin sensitivity, even if additional clinical studies are required to corroborate these assumptions.

Overall, I think that the manuscript is well-structured (within the scope of "Pharmaceuticals”) and of clinical impact on a current topic of interest.

I have some small suggestion/curiosity to improve the quality of review.

1    1) The current literature demonstrates convincing associations between diet, food components and metabolic syndrome management, including skin diseases as acne vulgaris. Indeed, it has been suggested that the rational intake of nutraceuticals, particularly abundant in foods typical of Mediterranean style eating patterns could be very useful in the prevention and cure of metabolic syndrome (i.e., Marini, H.R. Nutrients, 2022, 14, 1550).  Then, in light of the results here obtained, please to discuss on the possible co-application of nutraceutics and metformin that, in combination with healthy diet and physical activity, could provide a possible further strategy to prevent complications and delay the progression of Noncommunicable diseases, also ameliorating the skin diseases.

2 2) Gut microbiota composition is related to dietary habit as well as to pathophysiology of diseases, also modulating the therapeutic effects of diet/nutraceuticals/drugs etc. Please discuss this very intriguing topic of current research in the revised version of the present narrative review.

Author Response

1) We thank the reviewer for the constructive comments. As requested, we provided to add the interesting article as a reference. The use of nutraceutics is indeed interesting to the treatment of acne, as the use of nutraceutics in other diseases, but we want it to focalize on the specific and potential use of metformin in the treatment of acne.

2) We thank you for your comment, we have added in our references a paper regarding the effect of diet on the pathogenesis of acne, including it during the text.

Reviewer 2 Report

The authors endeavour to write about the use of metformin in in the treatment of acne in men. This is a reasonable topic of discussion since it is seldom discussed in the literature.

There are a number of deficits in the production of the manuscript, which I will try to highlight

1. the formatting of the manuscript

The headings of the manuscript are Introduction > Insulin Resistance > Metformin >Discussion > conclusion. I would adhere to the generally established formatting guidlines of Introduction > Methods > results > discussion/conclusion. The methodology of the literature review for example is in the discussion section which is somewhat confusing. The search strategy lacks Boolean operators and a PRISMA diagram. The abstracts lacks a statement identifying the work as a review.

The manuscript is very heavy on the introduction, and is more a review on the pathophysiologic aspects of acne with a focus on the role of androgens and insulin resistance. I believe the reason for this is the lack of data pertaining to the use of metformin for the treatment of acne in men. The novelty of the article may lie from drawing data from the use of metformin in other "acneiform" conditions such as hidradenitis suppurativa (for example). An analysis of current british/american/european guidlines could have also been performed.

Extensive revision of grammar and synthax is required.

Author Response

We thank you for your comments; the association between acne and insulin resistance has not been investigated as thoroughly in males as in women, despite the fact that in adult men, acne prevalence has grown. Regarding the methodology of the literature review, we didn’t insert a PRISMA diagram because there have been a few articles in the last years regarding the specific use of metformin in men that are affected by acne.

We thank you for your comments; yes, the headings of the manuscript are Introduction > Insulin Resistance > Metformin >Discussion > conclusion. The reason because the mechanism of insulin resistance and the link of metformin in the incidence of acne; we couldn’t consider methods or results, as you suggested, regarding the lack of specific studies in men.

We thank you for your comment; the search for publications was conducted in September 2022. Candidate studies were retrieved from the PubMed database(http://www.ncbi.nlm.nih.gov) using the keywords acne; insulin resistance; metformin, and men. All papers written in English and published during the period 2002-2021 were reviewed. The reference list of all retrieved articles was also reviewed to identify additional eligible studies that were not indexed in the above database. Since it is only them, we want to underline with our contribution how underestimated acne in men is in research and the use of metformin in them, as affected in women.

We thank you for your comment. Yes, we agree regarding the introduction and the background of insulin resistance in acne because we want to highlight the connection and the interplay of insulin resistance in acne. We thank you, indeed, the reason for this is the lack of data pertaining to the use of metformin for the treatment of acne in men.

We provided to fix the English as requested with the help of Dr. Rojin Tajmir, who is an English native speaker. Please find the correction with auto track of Microsoft word and the comments on the modification.

Round 2

Reviewer 2 Report

In this resubmission the authors have made minor modifications to the text and included a search strategy.

It is the opinion of this reviewer that given the paucity of data the manuscript may better be reformatted as a letter.

Author Response

We thank you for your comments; nowadays, the association between acne and insulin resistance has not been investigated as thoroughly in males as in women, despite the fact that in adult men, acne prevalence has grown through the last years. The purpose of our review is to increase the interest of our colleagues and include also men affected with acne and insulin resistance and increase the studies that are present in the literature.

Round 3

Reviewer 2 Report

The authors recommend that the review should be considered as a way to stimulate interest in the role of metformin in the treatment of acne in Males.

Author Response

We thank you for your comment and appreciate that we want to stimulate the use of metformin in treating acne in males and encourage new studies.